# Comparing Close-Field and Open-Field Autorefractometry and Subjective Refraction

**DOI:** 10.3390/jcm14165680

**Published:** 2025-08-11

**Authors:** Veronica Noya-Padin, Noelia Nores-Palmas, Belen Sabucedo-Villamarin, Maria J. Giraldez, Eva Yebra-Pimentel, Hugo Pena-Verdeal

**Affiliations:** 1Applied Physics Department (Optometry Area), Facultade de Óptica e Optometría, Universidade de Santiago de Compostela, 15705 Santiago de Compostela, Spain; veronicanoya.padin@usc.es (V.N.-P.); belen.sabucedo@rai.usc.es (B.S.-V.); hugo.pena.verdeal@usc.es (H.P.-V.); 2Optometry Group, Instituto de Investigación Sanitaria Santiago de Compostela (IDIS), 15706 Santiago de Compostela, Spain

**Keywords:** autorefraction, NVision-K 5001, refractive error measurement, subjective refraction, Visionix VX120

## Abstract

**Background/Objectives**: Autorefractometers are valuable tools in clinical practice, but their accuracy is often questioned, especially in the pediatric population. This study aimed to compare refraction data from open-field and close-field autorefractometers and subjective refraction without using cycloplegia. **Methods**: A total of 50 eyes of 50 participants (19 males and 31 females, 11.8 ± 1.56 years) were evaluated. In a single visit, objective refraction was performed using NVision-K 5001 (open-field) and Visionix VX120 (close-field) autorefractometers, and subjective refraction using the fogging technique. Differences between procedures were assessed for sphere, spherical equivalent, and cylindrical vectors J_0_ and J_45_ using the Friedman test, followed by the post hoc Wilcoxon test as needed. **Results**: Significant differences were found in the sphere between the three procedures (all *p* ≤ 0.032). For the spherical equivalent, the Visionix VX120 differed significantly with the other two techniques (both *p* < 0.001), whereas no significant differences were found between NVision-K 5001 and subjective refraction (*p* = 0.193). Finally, no significant differences were observed for J_0_ and J_45_ vectors among the procedures (both *p* ≥ 0.166). **Conclusions**: There are certain discrepancies between autorefractometers and the subjective assessment of refractive error, most evident in measurements taken with the close-field device, possibly due to greater accommodative stimulation. However, in contexts such as visual screening or as a preliminary guide in the clinic, the values obtained by autorefractometry can provide useful information.

## 1. Introduction

Autorefractometers determine the refraction of the eye and provide an objective measurement of the refractive error of the eye. These devices are very useful tools in the daily clinical practice of optometrists and ophthalmologists, providing great assistance in guiding visual refraction [1]. In addition, the objective nature of the procedure and the quick measurement process prove these devices to be especially suitable for populations that may lose interest easily, such as paediatric participants [2]. Similarly, their ease of use makes these devices particularly useful for vision screening purposes, where quick, non-invasive measurements are essential. Moreover, these devices can be operated by diverse professionals, including not only optometrists and ophthalmologists but also general practitioners, paediatricians, or technicians [3]. Consequently, the applicability of autorefractometers for the detection of refractive errors in vision screening programmes has been expanded. Indeed, autorefractometers play a significant role in the detection of myopia in children. This refractive error is increasingly more widespread, and in the youngest children, it can be challenging to detect due to the lack of collaboration during other techniques with longer duration [2]. The early detection of this refractive defect is essential for prompt correction, as it tends to highly increase under uncorrected conditions [4], and it will also allow the necessary control strategies to be applied promptly [5].

However, the results of autorefractometers are often questioned, especially during the clinical assessment of children without the use of cycloplegia, as the ocular accommodation that is particularly active at this age can lead to measurement errors due to instrumental myopia induced by the device [2,6,7]. In order to reduce this effect, open-field autorefractometers were developed, in which the fixation card is actually at a distant point, as opposed to close-field models, in which the stimulus is close and just simulates the distance [8]. Even though these autorefractometers are designed to reduce the demand for accommodation by allowing fixation on a distant target, residual accommodation may still exist in children and adolescents.

In the context of visual screening, the routine use of cycloplegic agents is neither practical nor justified, as rapid and non-invasive procedures are required. Furthermore, many of the professionals involved in these tasks are not legally authorised to administer drugs. Even within the field of eye care professionals, the use of cycloplegic agents is legally restricted for optometrists in many countries, which limits their applicability in routine clinical and screening settings [9]. In this scenario, it is essential to assess whether autorefraction without cycloplegia can provide reliable data in children.

Therefore, the present study aimed to compare refraction data without cycloplegia of both types of devices (an open-field and a close-field autorefractometer) and to evaluate both against subjective refraction in a myopic paediatric population.

## 2. Materials and Methods

### 2.1. Sample and Inclusion Criteria

For the sample size calculation, the software PS Power and Sample Size Calculations Version 3.1.2 (Copyright by William D. Dupont and Walton D. Plummer) was used. With a standard deviation of the spherical equivalent of 0.93 D and a tolerance limit of 0.50 D [10,11], to have 80% power (type II error associated) for a significance level of α = 0.050 (type I error associated) with a conficende level of 95% to detect a clinical difference between values, the minimum number of subjects required was 30.

A total of 52 volunteer subjects (20 males and 32 females) with a mean age ± standard deviation of 11.7 ± 1.59 years (range from 8 to 14 years) attending the Centre’s Optometry Clinic for myopia assessment were recruited. Data were then selected for statistical analysis only from subjects who met the following inclusion criteria: being myopic, monocular Best Corrected Visual Acuity (BCVA) higher than 0.9 (decimal) on a Snellen Letter Chart at distance, no use of ocular medications or artificial tears, no use of medications that could significantly affect pupillary size or refractive status, and absence of irregular cornea or additional ocular pathology other than a refractive error [2,12]. Participants who had previously undergone eye surgery for any reason were also not included [2].

Informed consent was obtained from the legal guardians of each participant prior to inclusion in the study. The study protocol conformed to the ethical standards of the responsible committee on human experimentation (institutional and national) and with the Declaration of Helsinki. Approval for the study was obtained from the University Bioethics Committee (approval number: USC 04/2022).

### 2.2. Study Design

In a single-day visit, the refractive error was measured by three techniques. The order of the tests was determined by considering the potential for each technique to stimulate accommodation, from lowest to highest risk. Objective refraction by the NVision-K 5001 open-field autorefractometer was performed first; followed by subjective refraction performed monocularly using the fogging technique with BCVA recording; and finally, objective refraction by Visionix VX120 multi-diagnostic platform (close-field autorefraction). All refraction measurements were performed without the administration of cycloplegia. In addition, a detailed anamnesis was conducted to inquire about the presence of diseases and/or treatments that could influence the measurements, and an ocular topography.

#### 2.2.1. NVision-K 5001

The Shin-Nippon NVision-K 5001 (Rexxam Co., Kagawa, Japan) [8] is an open-field autorefractometer; this implies that the fixation target can be positioned at a distant location, thus minimizing the instrumental myopia. For the present study, the fixation target was at a distance of 4 m; thus, the device was adjusted to this value. Furthermore, the power accuracy of the device was set to 0.12 D, the vertex distance was set to 12 mm, and the auto-start function was selected. This function automatically performs 3 measurements when the focus is achieved.

#### 2.2.2. Visionix VX120

The Visionix VX120 (Visionix Luneau Technologies, Chartres, France) is a non-invasive multi-diagnostic optical platform that measures the anterior segment parameters of the eye and the objective refractive condition [13]. The device was used to measure the refractive error and to verify compliance with corneal regularity criteria by performing topography. The vertex distance was set at 12 mm, and the power accuracy of the device was 0.25 D. To achieve the desired focus, the VX-120 device incorporates an XYZ mechanism that automatically adjusts the device’s positioning to ensure precise alignment with the target area.

#### 2.2.3. Subjective Refraction

Subjective refraction was conducted with the fogging technique, which is usually recommended for the paediatric population. In this procedure, more positive spherical power is added to the objective refraction value obtained by the retinoscope, to bring the optical focus point before the retina (towards the vitreous). This creates an artificial myopia that induces the relaxation of accommodation [14]. For the present study, the fogging technique was performed monocularly by adding 2.00 D of additional positive power to the objective refractive value obtained by retinoscopy. The defogging process was performed in increments of 0.25 D in a stepwise continuation until the maximum plus lens to BCVA was reached. Astigmatism was checked using the clock dial chart and the Jackson cross-cylinder technique. Finally, the duochrome test was performed to check the accuracy of the refraction. The display of all optotypes was conducted on the 24-inch SMT4V OptoTab Polar screen [15].

### 2.3. Statistical Analysis

The measurements were conducted in both eyes of all subjects, but only one eye per participant was included in the analysis to avoid statistical dependence between the eyes of the same subject [16]. The selection of the eye was determined using an alternating right–left sequence across participants (ratio 1:1).

The refractive values obtained were recorded in spherocylindrical form with a negative cylinder. From these values, the spherical equivalent (SE) and Jackson cross-cylinders vectors at 0° (J_0_) and 45° (J_45_) were calculated for each procedure as follows [17]:
SE = Sphere power + (Cylinder power/2),(1)
J_0_ vector = −(Cylinder power/2) × cos (2 × Axis of cylinder),(2)
J_45_ vector = −(Cylinder power/2) × sin (2 × Axis of cylinder),(3)


Subsequent to the transformation of the data into spherical components and vectors, the sphere, the SE, and J_0_ and J_45_ vectors were imported to the SPSS statistical software v.25.0 for Windows (SPSS Inc., Chicago, IL, USA) for data analysis. The significance level was set at *p* ≤ 0.050 for all analyses. Prior to conducting the analyses, the normal distribution of the data was assessed using the Shapiro–Wilk test [18]. The vectors derived from the Visionix VX120 outcomes exhibited a normal distribution (Shapiro–Wilk, both *p* ≥ 0.157), while all other parameters showed a non-normal distribution (Shapiro–Wilk, all *p* ≤ 0.018); therefore, non-parametric tests were applied. Differences between the three refraction procedures for the sphere, SE, and cylindrical vectors were evaluated with the Friedman test, followed by the Wilcoxon signed-rank test for pairwise comparisons where applicable [18]. In order to avoid type I errors derived from multiple comparisons, the Bonferroni correction was applied, adjusting the final significance level by the number of comparisons [19]. For graphical purposes, Bland–Altman procedures were used for variables that demonstrated significant discrepancies in the Friedman analysis. This method describes the correlation or similarity between two variables, representing averages versus differences. The 95% Limits of Agreement (95% LoAs) were calculated as mean difference ± 1.96 × standard deviation [20].

## 3. Results

Based on the inclusion criteria, the final sample consisted of 50 eyes of 50 participants (19 males and 31 females) with a mean age of 11.8 ± 1.56 years in a range from 8 to 14 years. Two subjects were excluded as the BCVA of 0.9 was not achieved. The descriptive data of the final sample are shown in Table 1.

### 3.1. Differences Between Refractive Procedures for Sphere Value

Significant differences were found between the results of the three procedures for the sphere value (Friedman test; *p* < 0.001) (Table 1). When applying pairwise comparison, all procedures showed significantly different results from each other (Wilcoxon test; all *p* ≤ 0.032) (Table 2). The Bland and Altman plot of means versus differences between techniques for the sphere value is shown in Figure 1. A systematic bias is observed when NVision-K 5001 is compared against subjective refraction, with the autorefractometer tending to yield slightly more positive measurements (mean difference value of 0.12 D; range between 95% LoAs: 1.00 D) (Figure 1a). When comparing the Visionix VX120 with the subjective refraction, the opposite is observed. There is a systematic bias, although this is slightly more negative in the Visionix VX120 (a mean difference value of −0.13 D) (Figure 1b). The largest observed bias is between the NVision-K 5001 and Visionix VX120 measurements, with a mean difference value of 0.25 D. Similarly, the total range of variability is the widest of the three comparisons (range between 95% LoAs: 1.22 D), indicating a greater discrepancy between these two techniques compared to the other combinations (Figure 1c).

### 3.2. Differences Between Refractive Procedures for SE Value

An analysis of the differences based on the SE value found significant differences between the measurement techniques (Friedman test, *p* < 0.001) (Table 1). A pairwise comparison showed that the Visionix VX120 device reported significantly different data to NVision-K 5001 and subjective refraction (Wilcoxon test, both *p* ≤ 0.001), but no statistically different results were found between objective refraction measured with the NVision-K 5001 and subjective refraction (Wilcoxon test, *p* = 0.193) (Table 2). The Bland and Altman plot of means versus differences between techniques for the SE value is shown in Figure 1. The graph between NVision-K 5001 measurements and subjective refraction corroborates the findings of the Wilcoxon post hoc test, with the values clustering around the 0 D difference value (mean difference value of 0.07 D) (Figure 1d). The comparison of Visionix VX120 with subjective refraction reveals a tendency for the former to yield more negative measurements, with a mean difference of −0.26 D. Moreover, for the SE, this comparison shows the widest range within the 95% LoAs, with a value of 1.30 D (Figure 1e). Once more, the most significant discrepancy is observed when comparing the NVision-K 5001 and Visionix VX120 measurements (mean difference value of 0.33 D; range of 95% LoAs: 1.17 D) (Figure 1f).

### 3.3. Differences Between Refractive Procedures for Jackson Cross-Cylinders at 0° and 45°

Comparisons of the vector components of the refraction found no significant differences between the measurement procedures for either the J_0_ vector or the J_45_ vector (Friedman test, both *p* ≥ 0.166) (Table 1).

## 4. Discussion

The paediatric population, whose accommodation is particularly active, requires conscientious subjective refraction with techniques such as fogging to ensure relaxation of accommodation [2]. However, this procedure can be slow and is highly dependent on the patient’s cooperation. Therefore, the development of devices that can obtain an accurate refractive value quickly and with minimal patient cooperation is of great interest [21]. The present study aimed to compare the results of two different devices for the measurement of refractive conditions (one open-field and one close-field) and evaluate both against the gold standard: subjective refraction. For this purpose, the refractive value was divided into four components: sphere, SE, and cylindrical vectors at 0 and 45°. The results found for the close-field device (Visionix VX120) are consistent with previous research findings, showing significant differences for the sphere and SE, most of which deviated towards a more negative value than that obtained by the subjective refraction [22,23]. Instrumental myopia, which can occur because the fixation target is inside the instrument, is often cited as the cause [23]. The use of cycloplegic agents would be necessary to prevent stimulation of accommodation. As shown in Figure 2, the distribution of the differences in the sphere and SE values obtained with the subjective refraction versus Visionix VX120 refraction exhibits a shift toward the negative side of the axis. As previously mentioned, this suggests that the Visionix VX120 may overestimate myopia, possibly due to accommodation stimulation. This contrasts with the observed for the NVision-K 5001, whose distribution of differences is centered around zero value, suggesting an absence of systematic bias. While ±0.25 D traditionally represents the minimum perceptible refractive change, diverse studies indicate that differences within ±0.50 D are tolerable in clinical practice [24,25]. Furthermore, the meta-analysis by Wilson et al. [26] reported a mean difference of −0.55 D when comparing non-cycloplegic autorefraction with cycloplegic retinoscopy, highlighting the impact of accommodation in pediatric refraction. Although this discrepancy represents a systematic bias rather than a tolerance limit, it underscores the expected variability in non-cycloplegic settings, such as the one under investigation. For the Visionix VX120 device, 98% and 88% of the sphere and SE data, respectively, was fitted within this range (±0.50 D), while for the NVision-K 5001 device, it was 98% in both cases. Therefore, although the differences are statistically significant, from a clinical perspective, a substantial proportion of them fall within an acceptable tolerance range. This finding supports the usefulness of these devices in visual screening tasks and as a resource to guide visual examinations. However, since the Visionix VX120 sporadically exhibited discrepancies that exceeded the ±1.00 D threshold, individual interpretation of the results, particularly in close-field devices, should be approached with caution.

The NVision-K 5001 device showed significant differences with the other techniques in the sphere, but not in the SE when compared to subjective refraction. This could be indicative of several aspects related to the accuracy and contribution of the cylinder to the SE. For example, if NVision-K 5001 provides a slightly underestimated sphere value and a slightly overestimated cylinder value, the combination of both values in SE could operate as a compensation mechanism. The overall agreement and size of the error in the results obtained by this device are also consistent with previous research using other open-field autorefractometers [23,27]. Furthermore, Queiros et al. [28,29] have demonstrated that disparities between open-field autorefraction with and without cycloplegia can be regulated by employing a fogging technique to control accommodation. To accomplish this objective, a +2.00 D lens was positioned at a distance of 12 mm from the corneal vertex during the measurement process conducted with the autorefractometer. Subsequently, this value was subtracted from the final result obtained. Their observations revealed that there were no statistically significant differences between measurements obtained with cycloplegia and those taken with the +2.00 D lens. Given these observations, the use of this method may be a valuable strategy for improving the reliability of results by enhancing accommodative control. However, it should be applied with caution, as the accommodative response in the pediatric population is particularly active.

For cylinder vectors, studies show more divergent results, with some agreeing with the present study that there are no statistically significant differences between objective and subjective techniques [30], while others find statistical differences between procedures [27], or even differences for one vector but not for the other [31].

One strength of this study is its focus on a pediatric population, which is particularly intriguing given the highly active accommodative system in children. Additionally, the sample was exclusively composed of myopic children, ensuring that the refractive error did not confound the findings. However, this approach is also considered a limitation, as it restricts the generalization of the results to emmetropic and hyperopic populations. Further research is required to specifically evaluate the differences between procedures for pediatric emmetropic and hyperopic populations. A further limitation of the present study is the absence of cycloplegia. A comparison of the results obtained with open-field and close-field autorefractometers, as well as with subjective refraction, under both cycloplegic and non-cycloplegic conditions would be of clinical and scientific interest. Finally, another limitation of the present study is the potential variability between operators, particularly in subjective refraction. Although all procedures were performed in accordance with standardised protocols, slight variations in technique or interaction with pediatric participants may have introduced variability. Furthermore, although the order of the tests was designed to minimize accommodative stimulation, the lack of randomization and the absence of washout periods may have affected the results. In the interest of mitigating this potential bias, it is recommended that subsequent studies incorporate random sequences or rest periods between measurements.

## 5. Conclusions

The results show that close-field objective measuring devices (Visionix VX120) tend to provide more negative results, probably due to the instrumental myopia involved. Meanwhile, open-field objective devices (NVision-K 5001) display values that are more closely aligned with those obtained through subjective refraction. From a clinical perspective, these devices are useful in visual screening activities and as support for comprehensive eye evaluations.

## Figures and Tables

**Figure 1 jcm-14-05680-f001:**
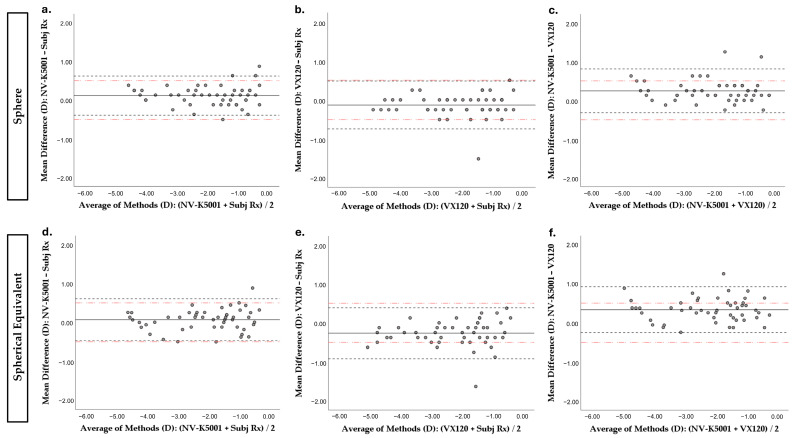
Mean versus differences (Bland–Altman plot) between spheric or SE refractive data between the different techniques. The solid horizontal line indicates the mean difference, the closely dashed horizontal lines indicate the 95% Limits of Agreement (mean difference ± 1.96 × standard deviation), and the red dash-dot horizontal lines indicate the clinically relevant limits (±0.50 D). (**a**) Spherical refractive error by NVision-K 5001 vs. spherical refractive error by subjective refraction; (**b**) spherical refractive error by Visionix VX120 vs. spherical refractive error by subjective refraction; (**c**) spherical refractive error by NVision-K 5001 vs. spherical refractive error by Visionix VX120; (**d**) SE by NVision-K 5001 vs. SE by subjective refraction; (**e**) SE by Visionix VX120 vs. SE by subjective refraction; and (**f**) SE by NVision-K 5001 vs. SE by Visionix VX120. D: Diopters. SE: spherical equivalent. NV-K5001: NVision-K5001. Subj Rx: subjective refraction. VX120: Visionix VX120.

**Figure 2 jcm-14-05680-f002:**
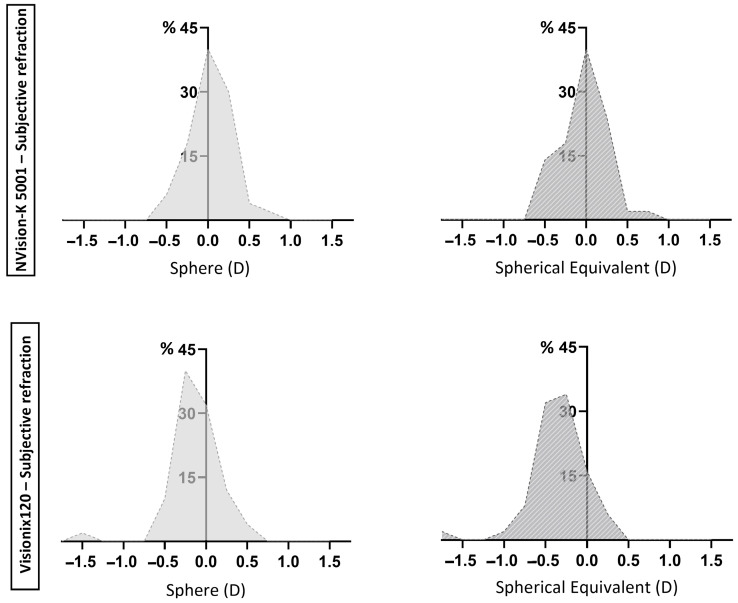
Percentage distribution of differences in sphere and spherical equivalent measurements from autorefractometers and subjective refraction. The upper section of the figure illustrates the distribution of differences between the NVision-K 5001 device and subjective refraction, whereas the lower section displays the distribution of differences between the Visionix VX120 device and subjective refraction.

**Table 1 jcm-14-05680-t001:** Descriptive statistics and differences between refractive components obtained by the three measurement methods. *n* = 50.

Parameter	Median [IQR]	Range	*p* ^a^
Sphere value (D)			<0.001 *
NV-K 5001	−1.62 [1.91]	−4.37 to 0.12	
VX120	−2.00 [2.06]	−5.00 to −0.25	
Subj Rx	−1.62 [1.87]	−4.75 to −0.25	
SE value (D)			<0.001 *
NV-K 5001	−1.71 [1.95]	−4.50 to −0.12	
VX120	−2.12 [1.91]	−5.38 to −0.38	
Subj Rx	−1.75 [1.72]	−4.75 to −0.50	
J_0_ vector			0.434
NV-K 5001	0.00 [0.30]	−0.68 to 0.62	
VX120	0.02 [0.23]	−0.44 to 0.46	
Subj Rx	0.00 [0.04]	−0.37 to 0.32	
J_45_ vector			0.166
NV-K 5001	0.00 [0.11]	−0.32 to 0.61	
VX120	−0.02 [0.34]	−0.48 to 0.36	
Subj Rx	0.00 [0.10]	−0.25 to 0.25	

^a^ Friedman test, * *p* < 0.050 statistically significant. IQR: Interquartile Range. J_0_ vector: cylindrical vector at 0°. J_45_ vector: cylindrical vector at 45°. NV-K5001: NVision-K5001. SE: spherical equivalent. Subj Rx: subjective refraction. VX120: Visionix VX120.

**Table 2 jcm-14-05680-t002:** Pairwise comparison between the refractive components obtained by the three measurement methods.

Variable	*p*	Adjusted *p* ^a^
Sphere value		
VX120—Subj Rx	0.011	0.032 *
VX120—NV-K5001	<0.001	0.001 *
Subj Rx—NV-K5001	0.007	0.021 *
SE value		
VX120—Subj Rx	<0.001	0.001 *
VX120—NV-K5001	<0.001	0.001 *
Subj Rx—NV-K5001	0.064	0.193

^a^ Significance values (*p*) obtained using the Wilcoxon test have been adjusted according to the Bonferroni correction, * *p* < 0.050 statistically significant. SE: spherical equivalent. NV-K5001: NVision-K5001. Subj Rx: subjective refraction. VX120: Visionix VX120.

## Data Availability

Data is unavailable due to privacy restrictions.

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
