# Peer review of "Comparing Close-Field and Open-Field Autorefractometry and Subjective Refraction"

_jcm, 2025, doi:10.3390/jcm14165680_

Round 1
Reviewer 1 Report
Comments and Suggestions for Authors
The study compared the refractive results of two autorefractometers—one open-field (NVision-K 5001) and one closed-field (Visionix VX120)—with subjective refraction (fogging technique) in 50 myopic children, without the use of cycloplegia. The differences in sphere values, spherical equivalent (SE), and cylindrical vectors (J0, J45) were analyzed. The results showed that the NVision-K 5001 had greater agreement with subjective refraction, while the VX120 showed a more negative bias, probably due to instrumental myopia. There was no significant difference in cylindrical vectors. The study suggests that open-field devices are more accurate for measuring refraction in children without cycloplegia.
I would like to raise a major concern regarding the lack of cycloplegia in this study, particularly given the pediatric nature of the sample. It is well-established that children and adolescents have highly active accommodative systems, which can significantly distort objective and subjective refraction results when cycloplegia is not used. The authors acknowledge this in the introduction but do not sufficiently justify the decision to omit cycloplegia in the methodology.
This limitation becomes even more critical considering the Bland-Altman analysis, which shows differences reaching or exceeding 1.00 D between methods. These discrepancies are clinically significant and cannot be dismissed. Therefore, the conclusion that "The NVision-K 5001 shows refractive values close to those obtained by subjective refraction" may be misleading to readers, potentially suggesting that non-cycloplegic autorefraction could be valid in pediatric assessments — which is not supported by the broader clinical evidence.
Moreover, the lack of discussion on accommodative activity is a clear omission. There is no mention of how accommodation might have influenced results, nor any attempt to quantify or control for its potential effects. This weakens the interpretability of the comparisons between devices.
A further methodological issue is the use of both eyes of each participant to effectively double the sample size. This approach violates the assumption of statistical independence, particularly in children where accommodation is often symmetrical and simultaneously stimulated. Since the measurements were taken consecutively, it's very likely that inter-eye correlation reduced variability artificially — yet this is not discussed or accounted for in the statistical analysis.
In light of these limitations, I strongly recommend that the authors reconsider their conclusions, provide a clearer justification for not using cycloplegia, and discuss the implications of accommodation and sample dependence on their results. Without this analysis, the findings risk misinterpretation in clinical practice.
Only after a thorough revision addressing these concerns and incorporating insights from these works would I be willing to proceed with a full review of the manuscript.
Author Response
The authors would like to thank the reviewer for taking the time to review this manuscript and for their suggestions to improve it. The comments have identified important areas that require clarification. Please find the detailed responses below and the corresponding corrections highlighted in red in the re-submitted files.
The study compared the refractive results of two autorefractometers—one open-field (NVision-K 5001) and one closed-field (Visionix VX120)—with subjective refraction (fogging technique) in 50 myopic children, without the use of cycloplegia. The differences in sphere values, spherical equivalent (SE), and cylindrical vectors (J0, J45) were analyzed. The results showed that the NVision-K 5001 had greater agreement with subjective refraction, while the VX120 showed a more negative bias, probably due to instrumental myopia. There was no significant difference in cylindrical vectors. The study suggests that open-field devices are more accurate for measuring refraction in children without cycloplegia.
Comment: I would like to raise a major concern regarding the lack of cycloplegia in this study, particularly given the pediatric nature of the sample. It is well-established that children and adolescents have highly active accommodative systems, which can significantly distort objective and subjective refraction results when cycloplegia is not used. The authors acknowledge this in the introduction but do not sufficiently justify the decision to omit cycloplegia in the methodology. This limitation becomes even more critical considering the Bland-Altman analysis, which shows differences reaching or exceeding 1.00 D between methods. These discrepancies are clinically significant and cannot be dismissed. Therefore, the conclusion that "The NVision-K 5001 shows refractive values close to those obtained by subjective refraction" may be misleading to readers, potentially suggesting that non-cycloplegic autorefraction could be valid in pediatric assessments — which is not supported by the broader clinical evidence.
Response: Thank you for the suggestions. The reviewer is right in highlighting that the absence of cycloplegic agents constitutes a limitation. The decision not to use pharmacological agents was based on two main considerations. First, these devices are often operated by a variety of healthcare professionals who may not have the legal authorization to administer medications. Furthermore, national regulations vary significantly, and in many countries, even optometrists are not permitted to use cycloplegic agents. Second, we were interested in evaluating the performance of these devices under non-cycloplegic conditions, as this more closely reflects their use in routine visual screening, where quick, non-invasive assessments are essential. This rationale has been further elaborated in the Introduction to justify the aims of the study better, and it is also addressed in the Discussion, particularly in the Limitations section, which has been substantially expanded. In addition, following the recommendations of reviewer 2, the tolerance limit has been revised, adjusting it to values that are more realistic in the clinical setting, particularly in pediatrics. Taking all of the above into account, the conclusions have been refined to cautiously reflect the potential utility of these measurements, positioning them primarily as a preliminary guide.
Comment: Moreover, the lack of discussion on accommodative activity is a clear omission. There is no mention of how accommodation might have influenced results, nor any attempt to quantify or control for its potential effects. This weakens the interpretability of the comparisons between devices.
Response: Thank you for pointing this out. The order of the tests was determined by considering the potential of the three techniques to stimulate accommodation, in an attempt to control the effect of accommodation on the results of this study. The methodology section has been revised to clarify this point. Furthermore, given the possibility that accommodation may have affected the results, both the order of the tests and the lack of cycloplegic use have been indicated as limitations of this study in the discussion section. Furthermore, the necessity of utilizing cycloplegic agents to avert accommodation stimulation in the event of instrumental myopia has been underscored in the discourse.
Comment: A further methodological issue is the use of both eyes of each participant to effectively double the sample size. This approach violates the assumption of statistical independence, particularly in children where accommodation is often symmetrical and simultaneously stimulated. Since the measurements were taken consecutively, it's very likely that inter-eye correlation reduced variability artificially — yet this is not discussed or accounted for in the statistical analysis.
Response: Thank you for the suggestion. While the original intention was to use both eyes of each participant for analysis, given that the focus was on a comparison between devices rather than between individuals, there is indeed a similarity in the accommodation activity of both eyes of an individual participant. Therefore, since the reviewer is correct, we have taken the decision to redo all statistical analyses using only one eye of each participant. The selection of the eye was determined using an alternating right-left sequence among participants. This has been detailed in the statistical analysis section of the methodology.
Comment: In light of these limitations, I strongly recommend that the authors reconsider their conclusions, provide a clearer justification for not using cycloplegia, and discuss the implications of accommodation and sample dependence on their results. Without this analysis, the findings risk misinterpretation in clinical practice. Only after a thorough revision addressing these concerns and incorporating insights from these works would I be willing to proceed with a full review of the manuscript.
Response: Thank you very much for your comments. The authors are grateful for the reviewer's thoughtful feedback and hope that the modifications made align with their expectations. As mentioned above, the decision not to use cycloplegic agents was based primarily on the legal regulation of drug use in each country, and also on the possibility of using these devices for visual screening. The objective was to explore the concordance between these measurement methods in scenarios where the use of cycloplegics is not feasible or not justified. The authors have attempted to provide a more thorough justification for this objective and address the issues that were identified.
Reviewer 2 Report
Comments and Suggestions for Authors
The authors conducted a prospective study comparing refractive data from an open-field autorefractor (NVision-K 5001), a closed-field autorefractor (Visionix VX120), and subjective refraction in a pediatric myopic population without cycloplegia. The manuscript addresses a clinically relevant topic in pediatric optometry, especially considering the increasing prevalence of myopia. The manuscript is generally well-written and methodologically sound. However, I have several comments regarding clarity, methodology, and clinical applicability.
Here are some comments to the authors:
1. The authors have not mentioned whether a power analysis or sample size calculation was conducted. Please clarify whether the sample size was predetermined to achieve adequate statistical power.
2. The testing order (open-field first, subjective second, closed-field last) may influence accommodation. Despite authors mentioning the rationale, a discussion on potential ordering bias and mitigation strategies (e.g., randomization or washout periods) would strengthen the study’s validity.
3. The study cites ±0.25 D as clinically acceptable variability. This threshold should be more thoroughly discussed and referenced, especially since some guidelines consider ±0.50 D acceptable, particularly in pediatric refractive studies.
4. The authors acknowledge the focus on myopic children as a limitation. I recommend expanding on other limitations:
-
- Lack of cycloplegia.
- The exclusion of emmetropic and hyperopic children, limiting generalizability.
- Potential inter-operator variability in subjective refraction, especially with fogging technique.
5. The manuscript is well-written, though some minor language polishing (e.g., occasional grammatical issues, sentence flow) could improve readability.
6. There are inconsistent usages of terms such as “closed-field” and “close-field.” I suggest unifying the terminology throughout the manuscript.
7. The tables use abbreviations like SE, J0, J45, and IQR without in-table footnotes. For clarity and to enhance stand-alone readability, these abbreviations should be fully defined at the bottom of each table.
8. The figures, particularly the Bland-Altman plots, are appropriate for assessing agreement between refraction techniques. However, their current presentation can be improved for clarity and clinical relevance. I recommend the authors enhance axis labeling (e.g., explicitly stating “Mean Difference (D)” and “Average of Methods (D)”), expand figure legends to briefly explain key interpretations (e.g., presence of bias or agreement limits), and consider adding clinically relevant reference lines (e.g., ±0.25 D limits) to help clinicians quickly gauge practical significance.
9. Some references are textbook-style summaries (e.g., StatPearls) rather than peer-reviewed studies. Consider prioritizing high-quality clinical studies, particularly regarding pediatric refraction and open-field technology.
Author Response
Thank you very much for taking the time to review this manuscript. The authors consider that the contributions have been extremely valuable and have improved the quality of the manuscript, both scientifically and in terms of readability. Please find the detailed responses below and the corresponding corrections highlighted in red in the re-submitted files.
The authors conducted a prospective study comparing refractive data from an open-field autorefractor (NVision-K 5001), a closed-field autorefractor (Visionix VX120), and subjective refraction in a pediatric myopic population without cycloplegia. The manuscript addresses a clinically relevant topic in pediatric optometry, especially considering the increasing prevalence of myopia. The manuscript is generally well-written and methodologically sound. However, I have several comments regarding clarity, methodology, and clinical applicability.
- The authors have not mentioned whether a power analysis or sample size calculation was conducted. Please clarify whether the sample size was predetermined to achieve adequate statistical power.
Response: Thank you very much for the suggestion. Following the reviewer's recommendations, a paragraph referring to the calculation of the sample size has been added to the Materials and Methods section. Additionally, following the suggestions of Reviewer 1, all statistical analyses have been redone using only one eye per participant to avoid potential inter-eye correlation. Accordingly, the normality test was also adjusted to reflect the updated sample size, in order to enhance the accuracy of the results.
- The testing order (open-field first, subjective second, closed-field last) may influence accommodation. Despite authors mentioning the rationale, a discussion on potential ordering bias and mitigation strategies (e.g., randomization or washout periods) would strengthen the study’s validity.
Response: Thank you for raising that point. The methodology section has been slightly adjusted to provide a clearer explanation of the measurement order. Additionally, the authors have expanded the limitations section of the discussion to include the lack of randomization and rest periods as limitations of this study.
- The study cites ±0.25 D as clinically acceptable variability. This threshold should be more thoroughly discussed and referenced, especially since some guidelines consider ±0.50 D acceptable, particularly in pediatric refractive studies.
Response: The reviewer is right. A new review of the literature has been conducted, and although a variation of ± 0.25 D has traditionally represented the minimum perceptible refractive change, various studies indicate that in clinical practice, a variation of ± 0.50 D is considered acceptable. This has been addressed and adapted in the discussion. Thank you very much.
- The authors acknowledge the focus on myopic children as a limitation. I recommend expanding on other limitations:
- Lack of cycloplegia.
- The exclusion of emmetropic and hyperopic children, limiting generalizability.
- Potential inter-operator variability in subjective refraction, especially with fogging technique.
Response: Thank you for your feedback. The limitations section has been expanded to incorporate the points suggested by the reviewer.
- The manuscript is well-written, though some minor language polishing (e.g., occasional grammatical issues, sentence flow) could improve readability.
Response: Thank you very much for the suggestion. A linguistic review of this work has been carried out to improve its readability.
- There are inconsistent usages of terms such as “closed-field” and “close-field.” I suggest unifying the terminology throughout the manuscript.
Response: Thank you for highlighting this issue. The term "close-field" and others in similar situations have been modified to ensure consistency of terminology throughout the manuscript.
- The tables use abbreviations like SE, J0, J45, and IQR without in-table footnotes. For clarity and to enhance stand-alone readability, these abbreviations should be fully defined at the bottom of each table.
Response: The authors appreciate the reviewer's comment. The table titles are placed above the respective tables, and the abbreviations are defined at the bottom of each table. There may have been a formatting issue, or it may not have been displayed correctly in the template. Nevertheless, we will check both tables to ensure that the footnotes are properly formatted in the revised version.
- The figures, particularly the Bland-Altman plots, are appropriate for assessing agreement between refraction techniques. However, their current presentation can be improved for clarity and clinical relevance. I recommend the authors enhance axis labeling (e.g., explicitly stating “Mean Difference (D)” and “Average of Methods (D)”), expand figure legends to briefly explain key interpretations (e.g., presence of bias or agreement limits), and consider adding clinically relevant reference lines (e.g., ±0.25 D limits) to help clinicians quickly gauge practical significance.
Response: Thank you for the indications. Both figures have been modified to adapt them to the new sample. In addition, following the reviewer's advice, axis labels with the required information have been added to the Bland-Altman plots to facilitate understanding of each graph. Throughout the text, the information relating to these graphs has also been expanded for improved interpretation. Regarding the addition of lines at the tolerance limits, the authors have opted not to include additional reference lines to avoid overloading the graph and potentially hindering rather than helping interpretation. However, if the reviewer considers that the inclusion of clinically relevant thresholds would improve the figure’s utility, we would be happy to incorporate them in a future revision.
- Some references are textbook-style summaries (e.g., StatPearls) rather than peer-reviewed studies. Consider prioritizing high-quality clinical studies, particularly regarding pediatric refraction and open-field technology.
Response: Thank you very much for the suggestion. The cited bibliography has been reviewed to replace those references that do not meet the required standards.
Round 2
Reviewer 1 Report
Comments and Suggestions for Authors
Reviewer Comments – Manuscript JCM-3765310
I would like to thank the authors for their detailed responses and for the revisions made to the manuscript. The changes address the main concerns raised in the previous review, particularly:
-
The reanalysis using only one eye per participant to preserve sample independence;
-
The expanded discussion on accommodative activity and its potential impact on the results;
-
The reformulation of the conclusions, which now reflect greater caution regarding the clinical applicability of non-cycloplegic measurements.
However, I believe the scientific discussion would benefit from referencing two key studies that directly address the core issue of this manuscript: the influence of residual accommodation in non-cycloplegic refraction, even when fogging techniques or open-field autorefractors are used. I strongly recommend including the following references in the discussion:
-
Influence of fogging lenses and cycloplegia on open-field automatic refraction. Ophthalmic Physiol Opt. 2008;28(4):387–92. doi:10.1111/j.1475-1313.2008.00579.x
-
Influence of fogging lenses and cycloplegia on peripheral refraction. J Optom. 2009;2(2):83–89. doi:10.3921/joptom.2009.83
Both studies (that I found in my research, and they are even by the same authors and unique in fogging) are highly relevant to the topic at hand and would provide important theoretical support for the limitations acknowledged by the authors. They clearly demonstrate that even with fogging or distant fixation targets, accommodation may not be fully relaxed in children, and that cycloplegia remains the gold standard in pediatric refractive assessment.
Additionally, I encourage the authors to acknowledge more explicitly that the differences observed in the Bland-Altman analysis—some exceeding 1.00 D—are clinically significant, not merely statistical. While it is understandable to discuss acceptable tolerances in screening contexts, this must be balanced with evidence showing that inaccurate refraction in children may lead to under- or overcorrection with real consequences.
To conclude, I find the study’s aim — to evaluate device performance under realistic, non-cycloplegic conditions — valid and clinically relevant. However, for the discussion to be fully supported and scientifically grounded, the addition of the references above and a brief revision of the discussion accordingly are recommended. With these minor adjustments, the manuscript could be suitable for publication.
Author Response
The authors would like to express their gratitude to the reviewer for the constructive feedback and suggestions provided, which have significantly enhanced the clarity and quality of the manuscript. In relation to this new review, please find the detailed responses below and the corresponding corrections highlighted in red in the re-submitted files.
I would like to thank the authors for their detailed responses and for the revisions made to the manuscript. The changes address the main concerns raised in the previous review, particularly:
- The reanalysis using only one eye per participant to preserve sample independence;
- The expanded discussion on accommodative activity and its potential impact on the results;
- The reformulation of the conclusions, which now reflect greater caution regarding the clinical applicability of non-cycloplegic measurements.
However, I believe the scientific discussion would benefit from referencing two key studies that directly address the core issue of this manuscript: the influence of residual accommodation in non-cycloplegic refraction, even when fogging techniques or open-field autorefractors are used. I strongly recommend including the following references in the discussion:
- Influence of fogging lenses and cycloplegia on open-field automatic refraction. Ophthalmic Physiol Opt. 2008;28(4):387–92. doi:10.1111/j.1475-1313.2008.00579.x
- Influence of fogging lenses and cycloplegia on peripheral refraction. J Optom. 2009;2(2):83–89. doi:10.3921/joptom.2009.83
Both studies (that I found in my research, and they are even by the same authors and unique in fogging) are highly relevant to the topic at hand and would provide important theoretical support for the limitations acknowledged by the authors. They clearly demonstrate that even with fogging or distant fixation targets, accommodation may not be fully relaxed in children, and that cycloplegia remains the gold standard in pediatric refractive assessment.
Response: Thank you for the indication, both studies have been referenced in the discussion section. The methodology employed by Queirós et al. has been detailed in the text, emphasizing its capacity to enhance the reliability of the outcomes by controlling accommodation. However, it is important to interpret the results with caution, especially in the pediatric population, whose accommodation is very active.
Additionally, I encourage the authors to acknowledge more explicitly that the differences observed in the Bland-Altman analysis—some exceeding 1.00 D—are clinically significant, not merely statistical. While it is understandable to discuss acceptable tolerances in screening contexts, this must be balanced with evidence showing that inaccurate refraction in children may lead to under- or overcorrection with real consequences.
Response: Thank you very much for the suggestion. A comment has been added to the discussion section to reflect the reviewer's comments. In addition, to more clearly visualize which values fall outside the ±0.50 D threshold in the Bland Altman plots, we have added a horizontal line to each of the plots marking the clinically relevant limits (±0.50 D).
To conclude, I find the study’s aim — to evaluate device performance under realistic, non-cycloplegic conditions — valid and clinically relevant. However, for the discussion to be fully supported and scientifically grounded, the addition of the references above and a brief revision of the discussion accordingly are recommended. With these minor adjustments, the manuscript could be suitable for publication.
Response: The authors hope that the modifications made meet the reviewer's standards, and we would like to reiterate our gratitude for the constructive input provided during the review process.
Reviewer 2 Report
Comments and Suggestions for Authors
The authors have addressed the initial concerns comprehensively, and the revised manuscript shows substantial improvements in clarity, methodology, and clinical relevance. The addition of power analysis, refined statistical approach, and expanded discussion greatly enhance the scientific value of the study. While the decision to omit clinical threshold lines from the Bland-Altman plots is understandable, their inclusion (e.g., as subtle dashed lines) could enhance interpretability without compromising visual clarity.
Remaining Minor Point: Reference 13 and 14 are co-authored by the current authors. Although relevant, it's good practice to ensure citation diversity. This is not a major issue unless over-reliance is detected.
Author Response
The authors have addressed the initial concerns comprehensively, and the revised manuscript shows substantial improvements in clarity, methodology, and clinical relevance. The addition of power analysis, refined statistical approach, and expanded discussion greatly enhance the scientific value of the study.
Response: The authors would like to thank the reviewer for providing positive feedback and constructive suggestions, which have contributed greatly to improving the quality and clarity of the manuscript.
While the decision to omit clinical threshold lines from the Bland-Altman plots is understandable, their inclusion (e.g., as subtle dashed lines) could enhance interpretability without compromising visual clarity.
Remaining Minor Point: Reference 13 and 14 are co-authored by the current authors. Although relevant, it's good practice to ensure citation diversity. This is not a major issue unless over-reliance is detected.
Response: Thank you for the indications. The clinically relevant limits have been incorporated into the Bland-Altman plots. To avoid any confusion with the existing elements of the figures, these lines have been added in red dash-dot style and this modification has been explicitly described in the corresponding figure legend. Furthermore, we have updated the reference list to ensure greater citation diversity and have removed the indicated references.
Once again, we would like to reiterate our gratitude for the constructive input provided during the review process.